# Biomarker panels for improved risk prediction and enhanced biological insights in patients with atrial fibrillation

Pascal B. Meyre [1,2] ✉, Stefanie Aeschbacher[1,2], Steffen Blum[1,2], Tobias Reichlin[3], Moa Haller[4,5], Nicolas Rodondi[4,5], Andreas S. Müller[6], Alain Bernheim[6], Jürg Hans Beer [7], Giorgio Moschovitis[8,9], André Ziegler[10], Bianca Wahrenberger[1,2], Elia Rigamonti[8,9], Giulio Conte[9], Philipp Krisai[1,2], Leo H. Bonati [11], Stefan Osswald[1,2], Michael Kühne[1,2] & David Conen[12,13,14]

Atrial fibrillation (AF) increases the risk of adverse cardiovascular events, yet the underlying biological mechanisms remain unclear. We evaluate a panel of 12 circulating biomarkers representing diverse pathophysiological pathways in 3817 AF patients to assess their association with adverse cardiovascular outcomes. We identify 5 biomarkers including D-dimer, growth differentiation factor 15 (GDF-15), interleukin-6 (IL-6), N-terminal pro-B-type natriuretic peptide (NT-proBNP), and high-sensitivity troponin T (hsTropT) that independently predict cardiovascular death, stroke, myocardial infarction, and systemic embolism, significantly enhancing predictive accuracy. Additionally, GDF-15, insulin-like growth factor-binding protein-7 (IGFBP-7), NT-proBNP, and hsTropT predict heart failure hospitalization, while GDF-15 and IL-6 are associated with major bleeding events. A biomarker model improves predictive accuracy for stroke and major bleeding compared to established clinical risk scores. Machine learning models incorporating these biomarkers demonstrate consistent improvements in risk stratification across most outcomes. In this work, we show that integrating biomarkers related to myocardial injury, inflammation, oxidative stress, and coagulation into both conventional and machine learning-based models refine prognosis and guide clinical decision-making in AF patients.

Patients with atrial fibrillation (AF) have an increased risk of death, cardiovascular events, and bleeding. Findings from prospective cohort studies suggest that AF is associated with an approximately 5-fold increased risk of stroke, 2-fold increased risk of myocardial infarction and death[1–3], and at least a 3-fold increased risk of heart failure[1,4]. Additionally, AF patients have an increased risk of major bleeding due to oral anticoagulation treatment[5]. As these complications carry significant risks for the individual patient and induce a significant burden to the healthcare system, it is of substantial clinical relevance to gain a better understanding of their underlying pathophysiology and to

improve risk prediction. Recent histological analyses from AF biopsies showed that the atrial tissue undergoes extensive changes over time, due to inflammation, blood clotting, vascular permeability, and collagen infiltration[6]. Circulating biomarkers that reflect these pathological pathways offer a promising and less invasive approach to assessing their involvement in individual patients, compared to more invasive diagnostic methods, such as tissue biopsies. Here, we report biomarker patterns associated with major adverse cardiac events and bleeding in AF patients, using a panel of selected biomarkers reflecting distinct disease pathways. By leveraging both traditional statistical

methods and machine learning models, we gain a deeper understanding of AF pathophysiology and improve risk prediction for major cardiovascular events and bleeding complications in this population.

## Results

### Baseline characteristics of the cohort

A total of 3817 AF patients were included in this analysis. Mean age was $71 \pm 10$ years, and 1067 (28%) were female (Table 1). Nearly half had paroxysmal AF (49%), and non-paroxysmal forms of AF were present in 51% (28% persistent and 23% permanent AF). Hypertension was the most common cardiovascular risk factor (69%), followed by coronary artery disease (27%) and heart failure (24%). Overall, 84% of patients were on oral anticoagulation. Spearman rank correlations between biomarkers are presented in Supplementary Fig. 1. HsTropT showed strong correlations (>0.60) with GDF-15 (0.64) and cystatin C (0.62). Osteopontin (OPN) was strongly correlated with cystatin C (0.77) and GDF-15 (0.64). IGFBP-7 exhibited strong correlations with cystatin C (0.68) and GDF-15 (0.67). NT-proBNP had a strong correlation with ANG-2 (0.69).

### Associations of biomarkers with cardiovascular outcomes

To identify biomarkers associated with cardiovascular events, we conducted age- and sex-adjusted, and multivariable-adjusted Cox regression analyses for each biomarker and outcome (Supplementary Tables 1–10 and in Supplementary Fig. 2). For the composite outcome of cardiovascular death, nonfatal ischemic stroke, nonfatal systemic embolism, or nonfatal myocardial infarction, 5 biomarkers including d-dimer, GDF-15, IL-6, NT-proBNP and hsTropT independently contributed to the model fit (Supplementary Table 1, Fig. 1A). Notably, hsTropT, NT-proBNP, and GDF-15 were among the most significant variables in the model (Fig. 2A). For heart failure hospitalization, 4 biomarkers - GDF-15, IGFBP-7, NT-proBNP and hsTropT - were significantly associated with the outcome (Supplementary Table 2, Fig. 1B), with NT-proBNP and GDF-15 being among the most important risk predictors (Fig. 2B). GDF-15, IGFBP-7, IL-6 and hsTropT were associated with an increased risk of major bleeding events (Supplementary Table 3, Fig. 1C), with GDF-15 being one of the most important biomarkers in the model (Fig. 2C). NT-proBNP and IL-6 were associated with both ischemic and the composite of ischemic and hemorrhagic stroke (Supplementary Tables 4 and 5, Fig. 1D, E), and NT-proBNP was among the most important risk indicators (Fig. 2D, E). Both IL-6 and hsTropT were linked to a higher risk of MI (Supplementary Table 6, Fig. 1F), and both were among the major predictors in the model (Fig. 2F). For cardiovascular death, GDF-15, IL-6, NT-proBNP, and hsTropT were associated with the outcome (Supplementary Table 7, Fig. 1G). GDF-15, NT-proBNP, and hsTropT were the most important predictors (Fig. 2G). D-dimer, GDF-15, IGFBP-7, IL-6, NT-proBNP, and hsTropT were associated with all-cause death (Supplementary Table 8, Fig. 1H). GDF-15, IL-6, and hsTropT were key risk predictors (Fig. 2H). For the composite bleeding outcome and clinically relevant NM bleeding, GDF-15, IL-6 for composite bleeding, and NT-proBNP for clinically relevant NM bleeding were associated with the outcomes (Supplementary Tables 9 and 10, Fig. 1I and J). GDF-15 was the most important risk indicator in the model (Fig. 2I, J). In sensitivity analyses restricted to patients on oral anticoagulation at baseline, the associations between biomarkers and major bleeding, all strokes, and ischemic stroke remained consistent (Supplementary Tables 11–13 and Supplementary Figs. 3 and 4).

### Comparison of biomarker model with clinical risk scores for stroke and major bleeding prediction

Figure 3 shows the discriminatory performance of the Cox model with and without biomarkers compared to clinical risk scores. For the composite stroke outcome, the inclusion of biomarkers significantly improved risk prediction relative to the $CHA_2DS_2$-VASc (AUC: 0.69 vs. 0.64; $P = 0.0003$) and the ABC-stroke score (AUC: 0.69 vs. 0.68; $P = 0.02$). For ischemic stroke, the biomarker model improved risk prediction as compared to the $CHA_2DS_2$-VASc (AUC: 0.68 vs. 0.63; $P = 0.003$) and the ABC-stroke score (AUC: 0.68 vs. 0.66; $P = 0.03$). For major bleeding, the biomarker model demonstrated improved predictive ability compared to the HAS-BLED score (AUC: 0.69 vs. 0.59; $P = 0.007$)

### Comparison of Cox and machine learning models with and without biomarkers for predicting cardiovascular outcomes

We investigated the impact of adding biomarkers to traditional Cox models and machine learning algorithms on cardiovascular risk prediction (Fig. 4, Supplementary Table 14). For composite outcome, the inclusion of biomarkers significantly improved model performance, increasing the AUC of the combined Cox model from 0.74 to 0.77 ($P = 2.6 \times 10^{-8}$). The random forest model showed an increase in AUC from 0.74 to 0.75 ($P = 0.03$), while the XGBoost model demonstrated an improvement from 0.95 to 0.97 ($P = 0.0007345$). For heart failure hospitalization, the inclusion of biomarkers enhanced predictive accuracy across all models. The combined Cox model's AUC increased from 0.77 to 0.80 ($P = 5.5 \times 10^{-10}$), the LASSO model from 0.80 to 0.83 ($P = 0.04$), the random forest model from 0.77 to 0.80 ($P = 0.0002564$), and the XGBoost model from 0.96 to 0.98 ($P = 5.0 \times 10^{-6}$). For major bleeding events, the addition of biomarkers resulted in improvements in some models. The combined Cox model's AUC increased from 0.67 to 0.68 ($P = 0.01$), and the random forest model showed a small but non-significant increase from 0.63 to 0.65 ($P = 0.10$). However, the LASSO model did not show a significant change (AUC 0.69–0.70, $P = 0.50$). In contrast, the XGBoost model exhibited an improvement, with the AUC increasing from 0.94 to 0.97 ($P = 8.8 \times 10^{-5}$). Similar trends were observed for all secondary outcomes. When comparing biomarker-based Cox and machine learning models to established clinical risk scores, the Cox and most machine learning models demonstrated higher AUC values than the ABC-stroke and $CHA_2DS_2$-VASc for stroke prediction, and the HAS-BLED for major bleeding (Supplementary Fig. 5). In sensitivity analyses restricted to patients receiving oral anticoagulation, the results for stroke and major bleeding remained consistent, with most models incorporating biomarkers showing higher AUC values (Supplementary Table 15 and Supplementary Fig. 6).

## Discussion

In this cohort of 3817 well-phenotyped AF patients, we identified several biomarkers associated with adverse cardiovascular events, including markers of myocardial injury (hsTropT), inflammation (IL-6), oxidative stress (GDF-15), coagulation (d-dimer), and cardiac dysfunction (NT-proBNP, IGFBP-7). The integration of these biomarkers into both traditional and machine learning-based predictive models significantly improved risk prediction, providing a more comprehensive assessment of adverse cardiovascular outcomes in this population. The improvement in predictive power was modest for most analyses.

Our analysis identified 6 biomarkers independently associated with AF-related complications and bleeding. GDF-15, a member of the TGF-β superfamily induced in cardiomyocytes, plays a significant role in oxidative stress, inflammation, cardiac injury, and fibrosis[7]. Epidemiological studies suggest that elevated GDF-15 levels increase the risk of bleeding in patients with cardiovascular diseases or AF[8–10]. Our findings not only confirm this association but also highlight GDF-15 as a robust predictor of heart failure hospitalization, with its predictive strength comparable to NT-proBNP and exceeding that of IGFBP-7 (Supplementary Table 2)[11]. IL-6, a well-established pro-inflammatory cytokine, has been linked to the pathophysiology of cardiovascular disease, particularly in AF patients[12]. Our study expands on these findings by demonstrating an association between IL-6 levels and both

**Table 1 | Baseline characteristics**

| Characteristic | N = 3817 |
|---|---|
| Age, years | 71 ± 10 |
| Female sex | 1067 (28) |
| Body mass index, kg/m² | 27 ± 5 |
| Systolic blood pressure, mmHg | 134 ± 19 |
| Heart rate, bpm | 70 ± 17 |
| CHA$_2$DS$_2$-VASc score | 3.2 ± 1.7 |
| Smoking status | |
| Active | 298 (8) |
| Past | 1836 (48) |
| Never | 1673 (44) |
| Type of atrial fibrillation | |
| Paroxysmal | 1869 (49) |
| Persistent | 1070 (28) |
| Permanent | 875 (23) |
| Medical history | |
| Hypertension | 2635 (69) |
| Diabetes | 614 (16) |
| Prior stroke/TIA | 651 (17) |
| Coronary artery disease | 1021 (27) |
| Prior myocardial infarction | 564 (15) |
| Peripheral vascular disease | 281 (7) |
| Heart failure | 909 (24) |
| Major bleeding | 146 (4) |
| Chronic kidney disease | 708 (19) |
| Oral anticoagulation | 3212 (84) |
| Direct oral anticoagulants | 1346 (35) |
| Vitamin K antagonists | 1864 (49) |

Data are presented as means ± standard deviations or counts (percentages).
*TIA* transient ischemic attack.

major and any bleeding events, suggesting that systemic inflammation may play a role in disrupting coagulation and vascular permeability. Additionally, IL-6 was significantly associated with stroke outcomes, consistent with findings from Mendelian randomization studies identifying IL-6 as a causal mediator of ischemic stroke in non-AF populations[13]. Genetic studies also underscore the relationship between IL-6 and atherosclerosis[14,15], a key risk factor for stroke. These findings warrant further investigation into the causal mechanisms linking IL-6 to both ischemic stroke and bleeding, as well as its potential therapeutic implications.

Our study underscores the multifactorial nature of AF-related complications, with diverse pathophysiological pathways contributing to risk. Observational studies have shown that a multimarker approach significantly improves risk prediction in both cardiovascular disease and AF populations[16–19]. We demonstrated that incorporating key biomarkers into prediction models led to a modest but significant improvement in the discriminatory ability of Cox and most machine learning models. This supports the concept that a biomarker panel reflecting the diverse tissue changes seen in AF provides a valuable approach for comprehensive cardiovascular risk assessment.

Emerging candidate biomarkers may capture additional biological aspects of AF-related outcomes. For example, the thrombin–antithrombin complex has been associated with worse outcomes in anticoagulated Asian patients with AF[20], while factor VIII antigen independently predicts stroke risk[21]. Bone morphogenetic protein 10 (BMP10) has been associated with incident AF in a population free of AF at baseline[22]. Moreover, among patients with established AF, elevated BMP10 levels have been linked to a higher risk of ischemic stroke independent of oral anticoagulation treatment[23], and an increased incidence of adverse outcome events compared to those with lower levels[24]. Future studies should investigate whether a more comprehensive proteomic analysis can provide deeper insights into the pathophysiology of AF complications and enhance risk stratification strategies.

Targeting multiple pathophysiological systems is essential for improving outcomes in complex cardiovascular conditions. For example, the polypill strategy has shown promise in stable coronary artery disease by simultaneously addressing multiple risk factors[25]. In AF patients, targeted treatments aimed at underlying cardiovascular conditions have been shown to improve sinus rhythm maintenance in persistent AF patients[26]; however, the impact of this strategy in reducing cardiovascular outcomes remains unclear. To date, no randomized trial has specifically evaluated the effects of such a multifaceted treatment approach in AF patients. Future clinical trials should therefore investigate whether comprehensive strategies targeting inflammation, coagulation disturbances, and cardiac dysfunction can improve long-term outcomes in this high-risk population.

The CHA$_2$DS$_2$-VASc score is a widely used tool for predicting stroke risk in AF patients[27]. However, its discriminatory performance is moderate at best[28]. Recent studies incorporating biomarkers, such as the ABC-stroke score, have demonstrated improved stroke prediction compared to the CHA$_2$DS$_2$-VASc score[29,30]. Our findings confirm the value of biomarkers in improving risk stratification in addition to clinical scores for both stroke and bleeding. The additive benefit of our biomarker panel was much larger when compared with biomarker-free scores (CHA$_2$DS$_2$-VASc, HAS-BLED) than compared to scores that already include some biomarkers (ABC-stroke), confirming that a biomarker-based approach strongly enhances risk prediction compared to models based exclusively on clinical variables. Further studies are needed to determine the optimal number of biomarkers for such models. Our data suggest that a more comprehensive biomarker-based model provides better risk prediction. Amplified P-wave duration has been associated with AF recurrence after ablation and worse prognosis[31,32]. Whether a combination with biomarkers further improves risk prediction in AF patients should be assessed in further studies.

We built machine learning models to leverage the entire set of clinical and biomarker variables for risk prediction. While traditional Cox regression models are limited by the number of predictors they can include without overfitting the model, machine learning models offer the advantage of capturing complex, non-linear relationships between clinical variables, biomarkers, and outcomes. In our study, we demonstrated that machine learning models, such as XGBoost and LASSO, achieved modest but significant improvements in predictive performance when biomarkers were included. These models, coupled with biomarker panels, have the potential to help clinicians identify patients who may benefit from further investigation and treatment. Future studies should assess whether the use of machine learning-based risk models can improve the management of AF patients.

This study has several limitations. First, the models were developed using a Swiss cohort, and their performance in external validation with other AF patient cohorts remains to be determined. However, we performed repeated cross-validation on the machine learning models, supporting the robustness of the results. Second, the study population was predominantly anticoagulated, which may limit the generalizability of our results to non-anticoagulated populations. Third, the absence of a non-AF comparison group limits our ability to determine whether the observed biomarker changes are specific to AF or linked to other pathophysiological processes. Future studies should incorporate appropriate control groups without AF or leverage Mendelian randomization analyses to improve our understanding of the pathophysiology of AF. Fourth, while our panel of biomarkers was

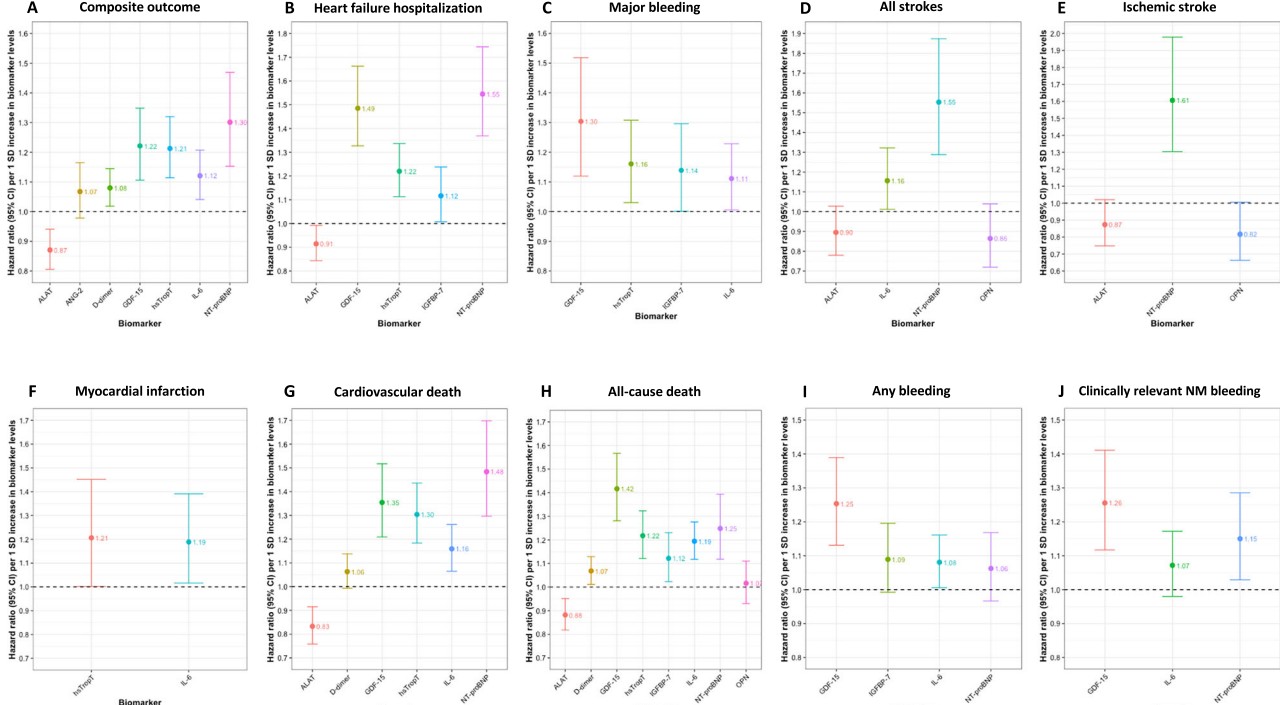

**Fig. 1 | Associations between selected biomarkers and adverse cardiovascular outcomes from combined Cox models.** This figure shows standardized hazard ratios and 95% CIs of associations between backward-selected biomarkers and different adverse cardiac outcomes, derived from combined multivariable Cox models. **A** shows effect estimates of selected biomarkers for composite outcome of cardiovascular death, nonfatal ischemic stroke, nonfatal systemic embolism, or nonfatal myocardial infarction. **B** shows effect estimates of selected biomarkers for heart failure hospitalization. **C** shows effect estimates of selected biomarkers for major bleeding. **D** shows effect estimates of selected biomarkers for all strokes. **E** shows effect estimates of selected biomarkers for ischemic stroke. **F** shows effect estimates of selected biomarkers for myocardial infarction. **G** shows effect estimates of selected biomarkers for cardiovascular death. **H** shows effect estimates of selected biomarkers for all-cause death. **I** shows effect estimates of selected biomarkers for any bleeding. **J** shows effect estimates of selected biomarkers for clinically relevant non-major (NM) bleeding. All outcomes were assessed in N = 3817 AF patients. Dots and whiskers represent hazard ratios and 95% CIs. ALAT Alanine aminotransferase, ANG-2 Angiopoetin-2, GDF-15 growth differentiation factor-15, hsTropT high-sensitivity troponin T, IGFBP-7 Insulin-like growth factor-binding protein-7, IL-6 Interleukin-6, NT-proBNP N-terminal pro-B-type natriuretic peptide, OPN Osteopontin.

chosen based on a thorough literature review and strong evidence linking them to AF pathophysiology and its complications, we acknowledge that any selection process is somewhat arbitrary and may miss relevant biomarker associations. Lastly, 16% of patients were not on oral anticoagulation, which may have influenced the associations between biomarkers and some outcomes. However, we showed that the results were consistent when analyses were restricted to those on anticoagulation.

In summary, our study highlights the complex, multifactorial nature of AF-related cardiovascular complications. We identified several biomarkers linked to diverse pathophysiological pathways - including myocardial injury, inflammation, oxidative stress, coagulation, and cardiac dysfunction - that are associated with adverse cardiovascular outcomes. By integrating these biomarkers into both traditional and machine learning-based risk models, we enhanced predictive accuracy, underscoring the potential clinical utility of biomarker-informed risk assessments in refining and optimizing the management of patients with AF.

## Methods

### Study population and procedures

We included patients with previously diagnosed AF from 2 prospective, multicenter cohort studies in Switzerland that used similar methodologies. The Basel Atrial Fibrillation (BEAT-AF) study enrolled 1545 patients from 2010 to 2014 across 9 centers in Switzerland[33], and the Swiss Atrial Fibrillation (Swiss-AF) study enrolled 2415 patients from 2014 to 2017 across 14 centers in Switzerland[34]. Both studies had

almost identical inclusion and exclusion criteria, as shown in Supplementary Table 16. Eligible patients had to have previously diagnosed AF. Patients who had secondary forms of AF or were unable to provide informed consent were excluded. For this analysis, we combined the BEAT-AF and Swiss-AF datasets, excluding 67 patients because of missing follow-up information and 76 patients because of missing all biomarker values, leaving a total of 3817 patients (Supplementary Fig. 7). Both studies comply with the Declaration of Helsinki, the study protocols were approved by the local ethics committees (Ethikkommission Nordwest- und Zentralschweiz (EKNZ)), and written informed consent was obtained from all participants. This study was conducted and reported in general accordance with the STARD guidelines[35].

At study enrolment and during yearly follow-up visits, trained study personnel collected information about patient demographics, risk factors, medical history, and current medical therapy using standardized case report forms. Sex of study participants was determined based on self-report. A detailed list of all variables collected is outlined in Supplementary Table 17. AF type was categorized according to guideline recommendations at the time of protocol development into paroxysmal, persistent, or permanent[36]. Body mass index was calculated as weight in kilograms divided by height in meters squared. Three consecutive blood pressure measurements were obtained at study enrolment, and the mean was used for all analyses. Estimated glomerular filtration rate (eGFR) was calculated using the Chronic Kidney Disease Epidemiology Collaboration (CKD-EPI) formula.

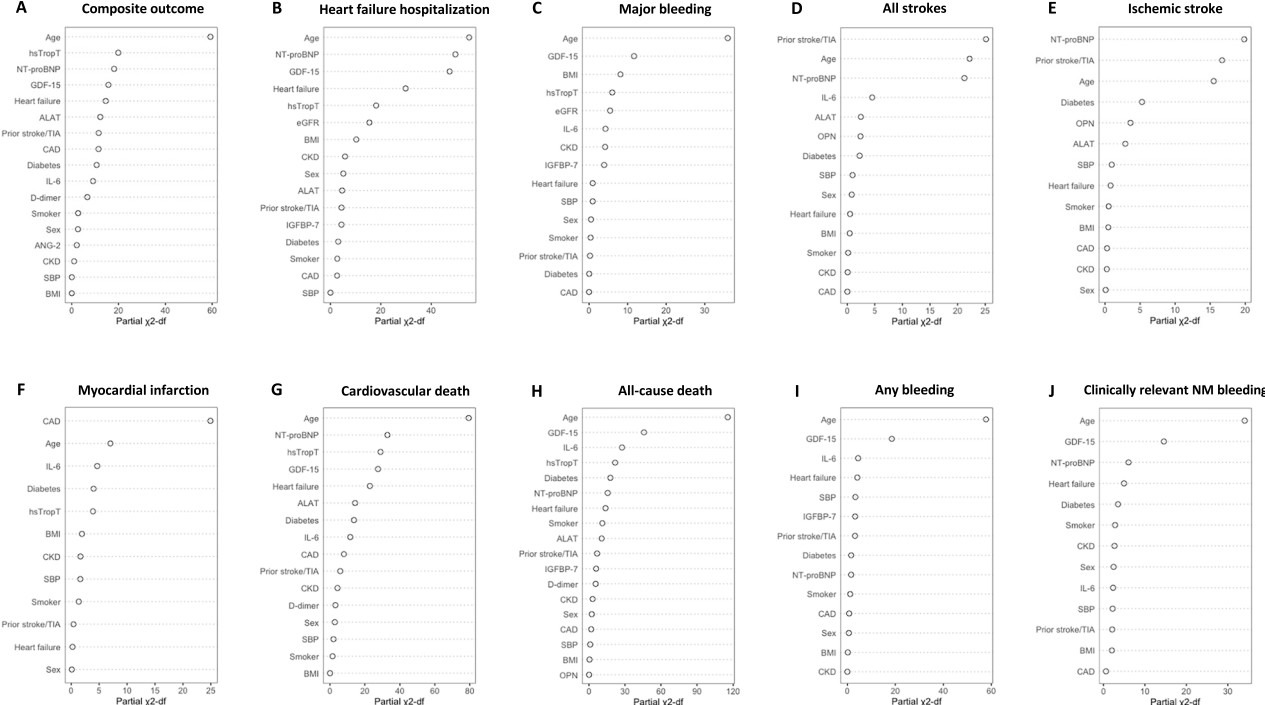

**Fig. 2 | Relative importance of predictors from combined Cox models.** This figure shows the relative importance of each clinical variable and backward-selected biomarkers for different adverse cardiac outcomes, derived from combined multivariable Cox models. **A** shows relative importance of variables for the association with the composite outcome of cardiovascular death, nonfatal ischemic stroke, nonfatal systemic embolism, or nonfatal myocardial infarction. **B** shows relative importance of variables for the association with heart failure hospitalization. **C** shows relative importance of variables for the association with major bleeding. **D** shows relative importance of variables for the association with all strokes. **E** shows relative importance of variables for the association with ischemic stroke. **F** shows relative importance of variables for the association with myocardial infarction. **G** shows relative importance of variables for the association with cardiovascular death. **H** shows relative importance of variables for the association with all-cause death. **I** shows the relative importance of variables for the association with any bleeding. **J** shows relative importance of variables for the association with clinically relevant non-major (NM) bleeding. All outcomes were assessed in $N = 3817$ AF patients. Dots represent the partial $\chi2$ – degree of freedom values. Source data are provided as a Source Data file. ALAT Alanine aminotransferase, ANG-2 Angiopoetin-2, BMI body mass index, CAD coronary artery disease, CKD chronic kidney disease, eGFR estimated glomerular filtration rate, GDF-15 growth differentiation factor-15, hsTropT high-sensitivity troponin T, IGFBP-7 Insulin-like growth factor-binding protein-7, IL-6 Interleukin-6, NT-proBNP N-terminal pro-B-type natriuretic peptide, OPN Osteopontin, SBP Systolic blood pressure.

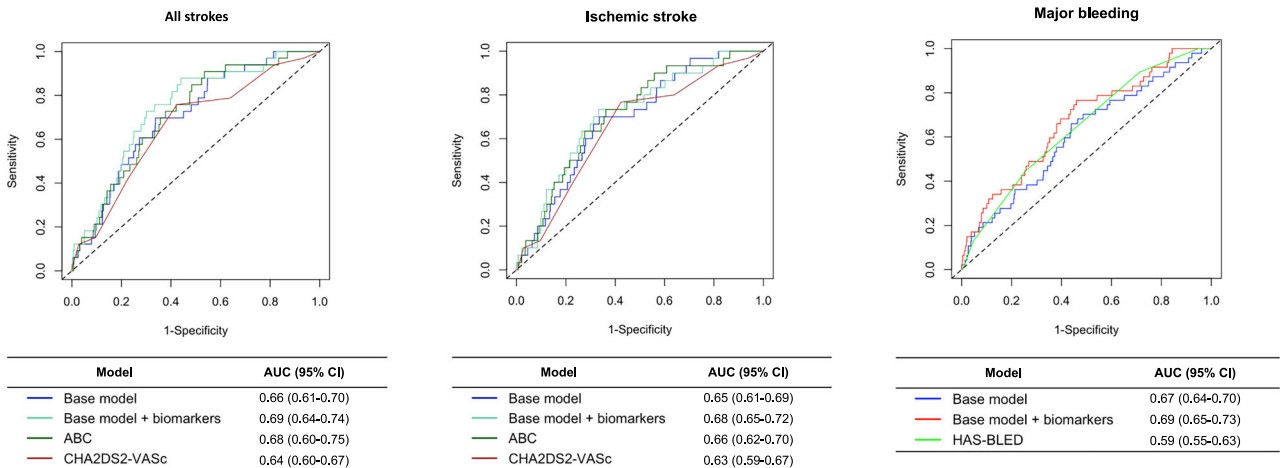

**Fig. 3 | Discriminatory performance of base Cox models with biomarkers vs. clinical scores for stroke and major bleeding.** This figure shows ROC curves of base Cox models with and without biomarkers and clinical risk scores for all strokes, ischemic stroke, and major bleeding. All outcomes were assessed in $N = 3817$ AF patients. ABC age, biomarkers, clinical history stroke risk score, CHA2DS2-VASc Congestive heart failure, Hypertension, Age (2 points if age >75 y), Diabetes, Stroke, Vascular disease, Sex category, HAS-BLED Hypertension, Abnormal renal/liver function, Stroke, Bleeding history or predisposition, Labile international normalized ratio, Elderly (>65 years), Drugs/alcohol concomitantly.

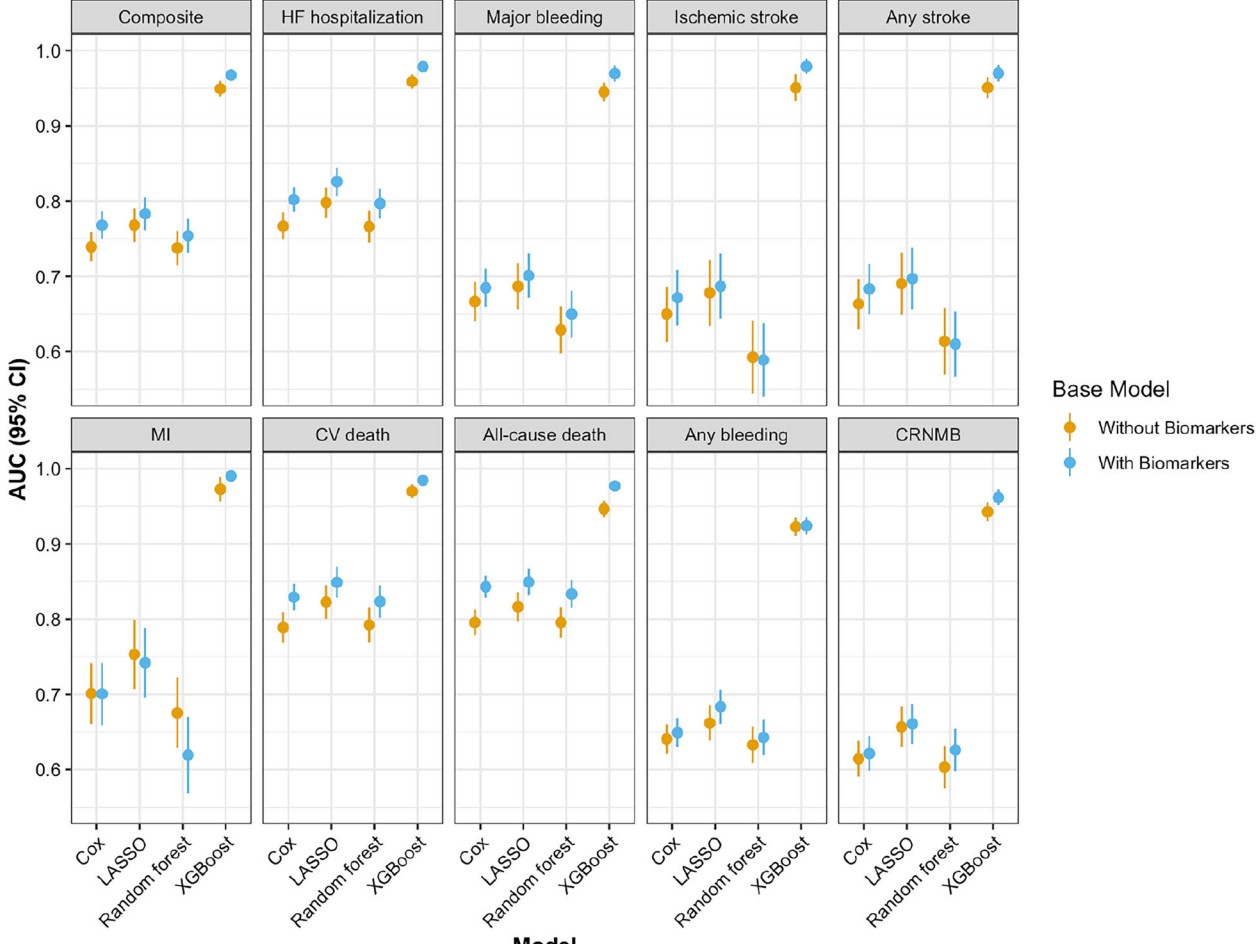

**Fig. 4 | Predictive performance of Cox and machine learning models for outcomes with and without biomarkers.** The figure shows the AUC with 95% CI for Cox and machine learning models, comparing the performance of base and base + biomarker models for different adverse cardiac outcomes. The combined Cox models include age, sex, body mass index, current smoker, systolic blood pressure, history of diabetes, prior stroke or TIA, history of heart failure, chronic kidney disease, coronary artery disease, and backward-selected biomarkers. The machine learning models include all variables listed in the Supplementary Table 16 and all biomarkers. All outcomes were assessed in $N = 3817$ AF patients. Dots represent AUC values and whiskers indicate 95% CIs. AUC area under the curve.

## Biomarker analyses and multiple imputation

Blood samples were drawn at baseline, immediately processed, and stored at −80 °C in a central biobank. We measured a panel of 12 biomarkers selected through an extensive literature review and robust evidence linking them to AF pathophysiology and its related complications. These biomarkers were chosen to capture distinct biological processes, including myocardial injury (hsTropT), inflammation (hs-CRP, IL-6, IGFBP-7, GDF-15), oxidative stress (GDF-15), renal disease (creatinine, cystatin C, OPN), coagulation (d-dimer), myocardial wall stress (NT-proBNP), extracellular matrix remodelling (IGFBP-7), liver disease (ALAT) and angiogenesis (ANG-2, IGFBP-7) (Supplementary Fig. 8). Biomarkers were analyzed centrally at Roche Diagnostics, Penzberg (Germany) on a cobas c311 or e601 by laboratory personnel blinded to clinical information under constant quality control and calibration. Most of the assays were routine products running on routine clinical analyzers. Detailed description about biomarker measurement is provided in the Supplementary Table 18.

To handle missing biomarker data in our dataset, we first assessed the percentage of missing values using a custom function pMiss, which calculates the proportion of missing values for each variable and each observation. The results showed that biomarkers had between 0.3% and 9.3% missing data. We then employed the *mice* package to visualize the missing data pattern and impute missing values using the predictive mean matching (PMM) method across 5 imputations. The imputed dataset was summarized and visually inspected using density plots and strip plots to assess the distribution and consistency of the imputed values (Supplementary Fig. 9). This approach ensures robust handling of missing data while preserving the integrity of the biomarker dataset for subsequent statistical analyses.

## Adverse cardiovascular outcome measures

The three main outcomes of this analysis were: (1) a composite of cardiovascular death, nonfatal ischemic stroke, nonfatal systemic embolism, or nonfatal myocardial infarction, (2) heart failure hospitalization, and (3) major bleeding. Secondary outcomes were the individual components of the composite outcome, as well as total stroke, myocardial infarction (MI), clinically relevant non-major (NM) bleeding, a composite of major or clinically relevant NM bleeding, and all-cause death. Definitions of all outcomes were identical in both cohorts, and definitions are provided in Supplementary Table 19. All clinical outcomes were adjudicated by blinded study personal and physicians.

## Development of machine learning models

To capture non-linear and complex relationships between clinical variables, biomarkers, and outcomes, different machine learning methods were applied. Specifically, 3 statistical methods including Least Absolute Shrinkage and Selection Operator (LASSO), random

forest, and Extreme Gradient Boosting for survival analysis (XGBoost)[37–39] were used for all outcomes. We applied three statistical methods to develop predictive models for primary and secondary outcomes. For this, we included the whole sample of clinical variables (46 variables detailed in Supplementary Table 17) and constructed two types of models: a base model incorporating only clinical variables, and an enhanced model that included both clinical variables and all biomarkers (base model + biomarkers). The machine learning techniques used to build these models are outlined below.

**Least absolute shrinkage and selection operator (LASSO).** LASSO was used to perform variable (feature) selection, regularization, and build parsimonious models for the primary and secondary outcomes. In brief, LASSO is a regularized regression analysis that simultaneously performs variable selection and regularization using machine learning algorithms. First, the feature matrix was prepared by converting the dataset into a model matrix excluding the intercept column. The response variable was set as the outcome being studied. LASSO logistic regression was then performed using the cv.glmnet function from the *glmnet* package, with 10-fold cross-validation to optimize the model's regularization parameter (lambda). The optimal lambda was determined by the value that minimized the cross-validated error (lambda.min). The resulting model coefficients were extracted, and non-zero coefficients were identified as the selected variables.

**Random forest.** Random forest, an ensemble learning method that enhances predictive accuracy by combining multiple decision trees, was used to develop models for predicting primary and secondary outcomes. Each variable was considered as a potential predictor, and the algorithm built an ensemble of decision trees, each trained on different subsets of variables. The final prediction was derived from the average of individual tree predictions. The random forest models were constructed using the *randomForest* package in R, with reproducibility ensured by setting a random seed. The models were trained with default settings. Predictive probabilities for the composite outcome were generated using the trained model, and model accuracy was evaluated using the confusionMatrix function. Additionally, model performance was assessed through 10-fold cross-validation across 100 decision trees.

**Extreme gradient boosting (XGBoost).** The XGBoost algorithm builds a series of models for the outcome being studied, each focusing on correcting residuals ("boosting") of the combined predictions of the models built so far. Specifically, each new model aims to capture the relationships in the data that were not well represented by the previous models. To develop predictive models using the XGBoost algorithm, we transformed outcome variables to a binary numeric format. The XGBoost model was trained on the dataset with a random seed set for reproducibility, using the *xgboost* package in R. The model was configured with a binary logistic objective and trained over 10 boosting rounds. Post-training, variable importance was assessed using the xgb.importance function, which ranked the features based on their contribution to the model. True outcomes were used to evaluate the model's performance, allowing for a robust assessment of its predictive capabilities.

**Statistical analyses**
The normality of distribution of each biomarker was assessed through visual inspection of histograms. Spearman rank correlations were applied to evaluate interrelationships between biomarkers. All biomarkers were log-transformed to improve the normality of the distribution. Cox proportional hazard models were constructed to assess hazard ratios (HR) and 95% confidence intervals (CI) for the main and secondary outcomes. To provide a unit-independent comparison between log-transformed biomarkers, HRs were standardized, representing the effect per 1 standard deviation (SD) increase. The initial

model was adjusted for age and sex, and the multivariable model was adjusted for a prespecified list of covariates, including age, sex, body mass index, current smoker, history of hypertension, history of diabetes, prior stroke, history of heart failure, chronic kidney disease, and coronary artery disease. First, separate models were constructed for each biomarker. We then constructed a combined multivariable model including all biomarkers in a single model and performed a backward selection of the biomarkers using the Akaike information criterion (AIC) to exclude biomarkers. To assess the consistency of our findings, we conducted a sensitivity analysis for stroke and major bleeding, restricting the cohort to patients who were on oral anticoagulation therapy at baseline. In addition, we measured each variable's relative importance in the models by calculating the partial $\chi 2$ statistic minus the predictor degrees of freedom. We then calculated the Area Under the Curve (AUC) of the Receiver Operating Characteristic (ROC) curve with corresponding 95% CI for each Cox model with and without key biomarkers and compared them using DeLong's test.

We constructed predictive machine learning models using the entire sample of clinical variables to construct 2 types of models: a base model that only includes clinical variables, and an enhanced model incorporating both clinical variables and all biomarkers. The model's discriminative ability was evaluated using AUC and corresponding 95% CI. AUC between base model and base model + biomarkers were compared using DeLong's tests. Finally, we assessed the predictive performance of the Cox and machine learning models with biomarkers by comparing their AUC to established clinical risk scores for stroke (ABC-stroke and $CHA_2DS_2$-VASc)[29,40] and for major bleeding (HAS-BLED)[41]. All statistical analyses were performed using R version 4.3.1. A *P*-value < 0.05 was considered statistically significant.

**Reporting summary**
Further information on research design is available in the Nature Portfolio Reporting Summary linked to this article.

## Data availability
The data supporting the findings from this study are available in the article and its supplementary information. Deidentified individual-level data supporting the results of the study are not publicly available to protect patient privacy and to comply with the informed consent signed by the participants. Deidentified individual-level data supporting the results of the study may be made available from the corresponding author (Pascal B. Meyre) upon reasonable request and approval by the study steering committee. Written access proposals have to be submitted to the corresponding author at the following e-mail address: pascal.meyre@usb.ch. The expected timeframe for responses is 10 weeks. Source data are provided with this paper.

## Code availability
The code is publicly available via GitHub−Zenodo and can be accessed using the https://doi.org/10.5281/zenodo.15653551. The source code from the R-packages used in this study is freely available online (https://cran.r-project.org/).

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

## Acknowledgements

This work was supported by grants of the Swiss National Science Foundation (grant numbers 33CS30_148474, 33CS30_177520, 32473B_176178, and 32003B_197524) awarded to D.C., M.K., and S.O.; the Swiss Heart Foundation, the Foundation for Cardiovascular Research Basel (FCVR), and the University of Basel awarded to D.C. The BEAT-AF study was supported by the Swiss National Science Foundation (grant number PP00P3_159322) awarded to D.C., the Swiss Heart Foundation,

the University of Basel, Boehringer Ingelheim, Sanofi-Aventis, Merck Sharp & Dome, Bayer, Daiichi Sankyo, and Pfizer/Bristol-Myers Squibb, all awarded to D.C., M.K., and S.O.

## Author contributions

P.B.M. and D.C. conceived and planned the study. P.B.M., A.Z. and D.C. performed the technical parts and analytic approach. P.B.M. and D.C. analyzed the data and contributed to the original interpretation. P.B.M. and D.C. wrote the draft manuscript, and S.A., S.B., T.R., M.H., N.R., A.S.M., A.B, J.H.B., G.M., A.Z., B.W., E.R., G.C., P.K., L.H.B., S.O., M.K. discussed the results and contributed to the final manuscript.

## Competing interests

P.B.M. received funding from the Swiss National Science Foundation outside the submitted work. S.A. received funding from the Swiss Heart Foundation and speaker fees from Roche Diagnostics outside of the submitted work. S.B. received funding from the Swiss National Science Foundation, the Mach-Gaensslen Foundation, and the Bangerter-Rhyner Foundation outside the submitted work. T.R. reports research grants from the Swiss National Science Foundation, the Swiss Heart Foundation, and the sitem insel support fund, all for work outside the submitted study. Speaker/consulting honoraria or travel support from Abbott/SJM, AstraZeneca, Brahms, Bayer, Biosense Webster, Biotronik, Boston Scientific, Daiichi Sankyo, Medtronic, Pfizer BMS, and Roche, all for work outside the submitted study. Support for his institution's fellowship program from Abbott/SJM, Biosense Webster, Biotronik, Boston Scientific, and Medtronic for work outside the submitted study. A.M. reports fellowship and training support from Biotronik, Boston Scientific, Medtronic, Abbott/St. Jude Medical, and Biosense Webster; speaker honoraria from Biosense Webster, Medtronic, Abbott/St. Jude Medical, AstraZeneca, Daiichi Sankyo, Biotronik, MicroPort, Novartis, and consultant honoraria for Biosense Webster, Medtronic, Abbott/St. Jude Medcal and Biotronik. G.M. has received consultant fees for taking part in advisory boards from Novartis, Boehringer Ingelheim, Bayer, AstraZeneca, and Daiichi Sankyo, all outside of the current work. A.Z. is an employee of Roche Diagnostics, a commercial provider of diagnostic tests. M.K. reports personal fees from Bayer, personal fees from Böhringer Ingelheim, personal fees from Pfizer BMS, personal fees from Daiichi Sankyo, personal fees from Medtronic, personal fees from Biotronik, personal fees from Boston Scientific, personal fees from Johnson&Johnson, grants from Bayer, grants from Pfizer, grants from Boston Scientific, grants from BMS, grants from Biotronik. Grants from the Swiss National Science Foundation, the Swiss Heart Foundation, the Foundation for Cardiovascular Research Basel, and the University of Basel. D.C. has received consultant fees from Roche Diagnostics and Trimedics, outside of the current work. The remaining authors have nothing to disclose.

## Additional information

[1]Department of Cardiology, University Heart Center, University Hospital Basel, Basel, Switzerland. [2]Cardiovascular Research Institute Basel, University Hospital Basel, Basel, Switzerland. [3]Department of Cardiology, Inselspital, Bern University Hospital, Bern, Switzerland. [4]Institute of Primary Health Care (BIHAM), University of Bern, Bern, Switzerland. [5]Department of General Internal Medicine, Inselspital, Bern University Hospital, University of Bern, Bern, Switzerland. [6]Department of Cardiology, Triemli Hospital Zurich, Zurich, Switzerland. [7]Department Internal Medicine, Baden Switzerland and Center of Molecular Cardiology, Cantonal Hospital Baden, University of Zürich, Zürich, Switzerland. [8]Divison of Cardiology, Regional Hospital of Lugano, Ente Ospedaliero Cantonale (EOS), Lugano, Switzerland. [9]Cardiocentro Ticino Institute, Ente Ospedaliero Cantonale (EOC), Lugano, Switzerland. [10]Roche Diagnostics International, Rotkreuz, Switzerland. [11]Rheinfelden Rehabilitation Clinic, Rheinfelden, Switzerland. [12]Population Health Research Institute, McMaster University, Hamilton, ON, Canada. [13]Department of Medicine, McMaster University, Hamilton, ON, Canada. [14]Department of Health Research Methods, Evidence, and Impact, McMaster University, Hamilton, ON, Canada. ✉e-mail: pascal.meyre@usb.ch

