## [Transparent Peer Review file · Nature Communications]

Biomarker panels for improved risk prediction and enhanced biological insights in patients with atrial fibrillation

Corresponding Author: Dr Pascal Meyre

Version 0:

Reviewer comments:

Reviewer #1

(Remarks to the Author)

Meyre et al. investigated a panel of 12 circulating biomarkers representing diverse pathophysiological pathways in a cohort of 3,817 AF patients to assess their association with adverse cardiovascular outcomes: all-cause death, cardiovascular death, total stroke, nonfatal ischemic stroke, nonfatal systemic embolism, nonfatal myocardial infarction, heart failure hospitalization, major bleeding, clinically relevant non-major (NM) bleeding, a composite of major or clinically relevant NM bleeding, and a composite of cardiovascular death, nonfatal ischemic stroke, nonfatal systemic embolism, or nonfatal myocardial infarction. They identified 5 biomarkers—d-dimer, growth differentiation factor 15 (GDF-15), interleukin-6 (IL-6), N-terminal pro-B-type natriuretic peptide (NT-proBNP), and high-sensitivity troponin T (hsTropT)—that were independently associated with cardiovascular death, stroke, myocardial infarction, and systemic embolism, significantly enhancing predictive accuracy. In addition, they exhibited that incorporating IL-6, NT-proBNP, and hsTropT to the CHA₂DS₂-VASc score improved stroke risk prediction. Furthermore, they demonstrated consistent improvements in risk stratification across all outcomes by incorporating these biomarkers using 3 machine learning models. The manuscript is well written, the methods and results are presented clearly and in sufficient detail and the conclusions in line with the findings of the study. However, several issues should be addressed.

Major concerns:

1. Selection of 12 biomarkers:

There are many biomarkers related to vascular changes, collagen infiltration, energy metabolism, inflammatory processes, myocardial wall tension and tissue injury. The rationale for selecting these 12 biomarkers out of other candidates should be provided.

2. Adjustment factors for each outcome:

Multivariable models for all outcomes are adjusted for age, sex, body mass index, current smoker, systolic blood pressure, history of diabetes, prior stroke or TIA, history of heart failure, chronic kidney disease and coronary artery disease. However, independent risk factors for each outcome are different. For example, anemia, left ventricular ejection fraction, and history of valvular heart disease have been shown to be independently associated with heart failure hospitalization and cardiovascular death.

Minor concerns:

P6, lines 133.

“the LASSO Model LASSO Model” should be corrected.

(Remarks on code availability)

Reviewer #3

(Remarks to the Author)

Meyre et al. present a study analyzing the predictive power of blood biomarkers for complications in atrial fibrillation patients.

Besides traditional risk models, they employ feature-based machine learning (LASSO, random forest, XGBoost). Adding biomarker results to the prediction models gave a statistically significant but modest (1-3%) increase in predictive power. For clinical translation, prospective trials will be required.

The work addresses a relevant question as such approaches could inform clinical decision-making to avoid complications in atrial fibrillation patients. My enthusiasm is dampened by the limited comparison to state of the art models. Meyre et al. compare to the established CHA2DS2-VASc score and a selected set of clinical variables (supplementary table 2). The study would benefit by broader comparison to other, more specific scores, such as HAS-BLED, ABC-stroke, amplified P-wave duration (<https://doi.org/10.1016/j.jacep.2017.12.001>).

The insight in the biological AF mechanisms underlying the reported complications would be even stronger if biomarker levels were compared against those in patients with the same clinical outcomes but without AF. In the current state it is hard to judge whether elevated/decreased biomarker levels are a cause of AF leading the outcome or linked to the outcome independent of the AF condition.

84% of the subjects were anticoagulated. Were results for stroke or bleeding different between coagulated and non-coagulated patients?

How were the biomarkers selected? I was surprised to see BMP10 neither included in the analysis nor mentioned in the discussion section (<https://doi.org/10.1093/eurheartj/ehae611>, <https://doi.org/10.1038/s41598-023-42331-7>).

The study demonstrates a modest but statistically significant added value of biomarkers for complication risk prediction. In my opinion a statement of the strength of the effect (modest) should be made as well wherever a statement about the statistical significance is made.

Minor points:

- line 66: "less invasive" compared to which other methods?
- line 89: the composite outcome should be defined in the main text
- line 153: please double check the sentence ("surpassing than")

(Remarks on code availability)

I appreciate that the code is available.

Ideally it would be persistently archived, for example via the GitHub2Zenodo integration.

The imputation script appears empty.

The other scripts seem solid at first glance except for some hard coded paths and the very long files without dedicated functions.

Version 1:

Reviewer comments:

Reviewer #1

(Remarks to the Author)

The authors appropriately corrected the manuscript. It is now acceptable for publication in Nature Communications.

(Remarks on code availability)

Reviewer #3

(Remarks to the Author)

The authors have improved the manuscript significantly and addressed all my comments sufficiently.

When looking at the code repository, it seems that the latest versions including the refactoring mentioned in the response to my comment has not been uploaded yet. Hard-coded paths and very long, rather unstructured files remain.

(Remarks on code availability)

When looking at the code repository, it seems that the latest versions including the refactoring mentioned in the response to my comment has not been uploaded yet. Hard-coded paths and very long, rather unstructured files remain.

Reviewer #1:

Meyre et al. investigated a panel of 12 circulating biomarkers representing diverse pathophysiological pathways in a cohort of 3,817 AF patients to assess their association with adverse cardiovascular outcomes: all-cause death, cardiovascular death, total stroke, nonfatal ischemic stroke, nonfatal systemic embolism, nonfatal myocardial infarction, heart failure hospitalization, major bleeding, clinically relevant non-major (NM) bleeding, a composite of major or clinically relevant NM bleeding, and a composite of cardiovascular death, nonfatal ischemic stroke, nonfatal systemic embolism, or nonfatal myocardial infarction. They identified 5 biomarkers—d-dimer, growth differentiation factor 15 (GDF-15), interleukin-6 (IL-6), N-terminal pro-B-type natriuretic peptide (NT-proBNP), and high-sensitivity troponin T (hsTropT)—that were independently associated with cardiovascular death, stroke, myocardial infarction, and systemic embolism, significantly enhancing predictive accuracy. In addition, they exhibited that incorporating IL-6, NT-proBNP, and hsTropT to the CHA₂DS₂-VASc score improved stroke risk prediction. Furthermore, they demonstrated consistent improvements in risk stratification across all outcomes by incorporating these biomarkers using 3 machine learning models. The manuscript is well written, the methods and results are presented clearly and in sufficient detail and the conclusions in line with the findings of the study. However, several issues should be addressed.

Response: We thank the reviewer for the interest in our work.

Major concerns:

1. Selection of 12 biomarkers:

There are many biomarkers related to vascular changes, collagen infiltration, energy metabolism, inflammatory processes, myocardial wall tension and tissue injury. The rationale for selecting these 12 biomarkers out of other candidates should be provided.

Response: The 12 biomarkers were chosen based on literature review and prior evidence linking them to key pathophysiological pathways implicated in AF and its complications. Specifically, these markers were selected to represent distinct biological pathways including myocardial injury (hsTropT), inflammation (hs-CRP, IL-6, IGFBP-7, GDF-15), oxidative stress (GDF-15), renal disease (creatinine, cystatin C, OPN), coagulation (d-dimer), myocardial wall stress (NT-proBNP), extracellular matrix remodelling (IGFBP-7), liver disease (ALAT) and angiogenesis (ANG-2, IGFBP-7). We have now expanded the methods section to provide a detailed rationale for our selection on page 23 as follows:

“We measured a panel of 12 biomarkers selected through an extensive literature review and robust evidence linking them to AF pathophysiology and its related complications. These biomarkers were chosen to capture distinct biological processes, including myocardial injury (hsTropT), inflammation (hs-CRP, IL-6, IGFBP-7, GDF-15), oxidative stress (GDF-15), renal disease (creatinine, cystatin C, OPN), coagulation (d-dimer), myocardial wall stress (NT-proBNP), extracellular matrix

remodelling (IGFBP-7), liver disease (ALAT) and angiogenesis (ANG-2, IGFBP-7) (Supplementary Figure 8).”

We added an additional figure in the Supplement for further illustration:

We also included the potential implications of other candidate markers in the discussion on page 9:

“Emerging candidate biomarkers may capture additional biological aspects of AF-related outcomes. For example, the thrombin–antithrombin complex has been associated with worse outcomes in anticoagulated Asian patients with AF²⁰, while factor VIII antigen independently predicts stroke risk²¹. Bone morphogenetic protein 10 (BMP10) has been associated with incident AF in a population free of AF at baseline²². Moreover, among patients with established AF, elevated BMP10 levels have been linked to a higher risk of ischemic stroke independent of oral anticoagulation treatment²³, and an increased incidence of adverse outcome events compared to those with lower levels²⁴. Future studies should investigate whether a more comprehensive proteomic analysis can provide deeper insights into the pathophysiology of AF complications and enhance risk stratification strategies.”

We added the following references:

- 20 Koretsune, Y. et al. Coagulation Biomarkers and Clinical Outcomes in Elderly Patients With Nonvalvular Atrial Fibrillation: ANAFIE Subcohort Study. *JACC Asia* 3, 595-607 (2023).
- 21 Singleton, M. J. et al. Multiple Blood Biomarkers and Stroke Risk in Atrial Fibrillation: The REGARDS Study. *J Am Heart Assoc* 10, e020157 (2021).
- 22 Chua, W. et al. An angiopoietin 2, FGF23, and BMP10 biomarker signature differentiates atrial fibrillation from other concomitant cardiovascular conditions. *Sci Rep* 13, 16743 (2023).
- 23 Hijazi, Z. et al. Bone morphogenetic protein 10: a novel risk marker of ischaemic stroke in patients with atrial fibrillation. *Eur Heart J* 44, 208-218 (2023).

24 Hennings, E. et al. Bone Morphogenetic Protein 10-A Novel Biomarker to Predict Adverse Outcomes in Patients With Atrial Fibrillation. J Am Heart Assoc 12, e028255 (2023).

We added the following statement in the limitations section about the selection process of biomarkers on page 11:

“Fourth, while our panel of biomarkers was chosen based on a thorough literature review and strong evidence linking them to AF pathophysiology and its complications, we acknowledge that any selection process is somewhat arbitrary and may miss relevant biomarker associations.”

2. Adjustment factors for each outcome:

Multivariable models for all outcomes are adjusted for age, sex, body mass index, current smoker, systolic blood pressure, history of diabetes, prior stroke or TIA, history of heart failure, chronic kidney disease and coronary artery disease. However, independent risk factors for each outcome are different. For example, anemia, left ventricular ejection fraction, and history of valvular heart disease have been shown to be independently associated with heart failure hospitalization and cardiovascular death.

Response: We agree that individual outcomes have specific individual risk factors. Our initial choice of adjustment variables was to ensure consistency across models while incorporating broadly relevant variables known to be widely available in clinical practice. Also, based on comments from reviewer #3 we now performed a series of benchmark analyses against known risk models, which in our view add more information than adding a few potential confounders to the individual models. Also in our dataset, specific information on left ventricular ejection fraction, anemia and valvular disease is missing in many patients, which precludes us from including them in the models. We hope the reviewer agrees with our decision.

Minor concerns:

P6, lines 133.

“the LASSO Model LASSO Model” should be corrected.

Response: We apologize for the typo and have corrected the error. The text now reads correctly on page 7 as follows:

“However, the LASSO model did not show a significant change (AUC 0.69 to 0.70, P=0.50).”

Reviewer #3:

Meyre et al. present a study analyzing the predictive power of blood biomarkers for complications in atrial fibrillation patients. Besides traditional risk models, they employ feature-based machine learning (LASSO, random forest, XGBoost). Adding biomarker results to the prediction models gave a statistically significant but modest (1-3%) increase in predictive power. For clinical translation, prospective trials will be required.

The work addresses a relevant question as such approaches could inform clinical decision-making to avoid complications in atrial fibrillation patients. My enthusiasm is dampened by the limited comparison to state of the art models. Meyre et al. compared to the established CHA2DS2-VASc score and a selected set of clinical variables (supplementary table 2). The study would benefit by broader comparison to other, more specific scores, such as HAS-BLED, ABC-stroke, amplified P-wave duration (<https://doi.org/10.1016/j.jacep.2017.12.001>).

Response: We thank the reviewer for the careful review of our manuscript.

We have now incorporated into our manuscript additional comparisons with established risk scores, including the ABC-stroke for all and ischemic stroke and HAS-BLED score for major bleeding. See the results below.

Figure 3. Discriminatory performance of base Cox model with biomarkers vs. clinical scores for stroke and major bleeding

Model	AUC (95% CI)
Base model	0.67 (0.64-0.70)
Base model + biomarkers	0.69 (0.65-0.73)
HAS-BLED	0.59 (0.55-0.63)

We then compared the discriminatory ability of clinical risk scores (CHA₂DS₂-VASc, ABC-stroke, HAS-BLED), base model + biomarkers and machine learning models + biomarkers in predicting all strokes, ischemic strokes, and major bleeding. The results are summarized below.

Supplementary Figure 5. Discriminatory performance of clinical risk scores, biomarker-based models, and machine learning models for predicting stroke and major bleeding

We updated the methods section in page 27 as follows:

“Finally, we assessed the predictive performance of the Cox and machine learning models with biomarkers by comparing their AUC to established clinical risk scores for stroke (ABC-stroke and CHA₂DS₂-VASC)^{29,39} and for major bleeding (HAS-BLED)⁴⁰.

We added the following references:

- 29 Hijazi, Z. et al. The ABC (age, biomarkers, clinical history) stroke risk score: a biomarker-based risk score for predicting stroke in atrial fibrillation. *Eur Heart J* 37, 1582-1590 (2016).
- 39 Lip, G. Y., Nieuwlaat, R., Pisters, R., Lane, D. A. & Crijns, H. J. Refining clinical risk stratification for predicting stroke and thromboembolism in atrial fibrillation using a novel risk factor-based approach: the euro heart survey on atrial fibrillation. *Chest* 137, 263-272 (2010).
- 40 Pisters, R. et al. A novel user-friendly score (HAS-BLED) to assess 1-year risk of major bleeding in patients with atrial fibrillation: the Euro Heart Survey. *Chest* 138, 1093-1100 (2010).

In the results section, we added the following statement on page 6:

“Figure 3 shows the discriminatory performance of the Cox model with and without biomarkers compared to clinical risk scores. For the composite stroke outcome, the inclusion of biomarkers significantly improved risk prediction relative to the CHA₂DS₂-VASC (AUC: 0.69 vs. 0.64; P=0.0003) and the ABC-stroke score (AUC: 0.69 vs. 0.68; P=0.02). For ischemic stroke, the biomarker model improved risk prediction as compared to the CHA₂DS₂-VASC (AUC: 0.68 vs. 0.63; P = 0.003) and the ABC-stroke score (AUC: 0.68 vs. 0.66; P = 0.03). For major bleeding, the biomarker model demonstrated improved predictive ability compared to the HAS-BLED score (AUC: 0.69 vs. 0.59; P = 0.007).”

We also included the following statement on page 7 regarding the comparison between Cox and machine learning biomarker models and clinical established risk scores for stroke and major bleeding:

“When comparing biomarker-based Cox and machine learning models to established clinical risk scores, the Cox and most machine learning models demonstrated higher AUC values than the ABC-stroke and CHA₂DS₂-VAsC for stroke prediction, and the HAS-BLED for major bleeding (Supplementary Figure 5).”

We rephrased the paragraph in the discussion section addressing the comparison of our models with clinical scores on page 10:

“The CHA₂DS₂-VAsC score is widely used tool for predicting stroke risk in AF patients²⁷. However, its discriminatory performance is moderate at best²⁸. Recent studies incorporating biomarkers, such as the ABC-stroke score, have demonstrated improved stroke prediction compared to the CHA₂DS₂-VAsC score^{29,30}. Our findings confirm the value of biomarkers in improving risk stratification in addition to clinical scores for both stroke and bleeding. The additive benefit of our biomarker panel was much larger when compared with biomarker-free scores (CHA₂DS₂-VAsC, HAS-BLED) than compared to scores that already include some biomarkers (ABC-stroke), confirming that a biomarker-based approach strongly enhances risk prediction compared to models based exclusively on clinical variables. Further studies are needed to determine the optimal number of biomarkers for such models. Our data suggest that a more comprehensive biomarker-based model provides better risk prediction.”

We updated the sentence in the abstract on page 3 as follows:

“A biomarker model improves predictive accuracy for stroke and major bleeding compared to established clinical risk scores.”

Although amplified P-wave duration is a known risk factor for incident AF and AF recurrence after ablation, we do not have these data available in our cohorts, and were therefore unable to include it in our analysis. We added a statement in the discussion section about amplified P-wave duration as potential risk modifier in AF patients on page 10:

“Amplified P-wave duration has been associated with AF recurrence after ablation and worse prognosis^{31,32}. Whether a combination of ECG and biomarkers data further improves risk prediction in AF patients should be assessed in further studies.”

We added the following references:

31 Jadidi, A. et al. The Duration of the Amplified Sinus-P-Wave Identifies Presence of Left Atrial Low Voltage Substrate and Predicts Outcome After Pulmonary Vein Isolation in Patients With Persistent Atrial Fibrillation. *JACC Clin Electrophysiol* 4, 531-543 (2018).

32 Magnani, J. W. et al. P wave duration is associated with cardiovascular and all-cause mortality outcomes: the National Health and Nutrition Examination Survey. *Heart Rhythm* 8, 93-100 (2011).

The insight in the biological AF mechanisms underlying the reported complications would be even stronger if biomarker levels were compared against those in patients with the same clinical outcomes but without AF. In the current state it is hard to judge whether elevated/decreased biomarker levels are a cause of AF leading the outcome or linked to the outcome independent of the AF condition.

Response: We agree that such comparisons could offer valuable mechanistic insights. However, for the main scope of risk prediction in the current study it is not that relevant whether there is a causal relationship between the biomarker and AF. However, we have included a statement acknowledging this limitation in the limitation section on page 11:

“Third, the absence of a non-AF comparison group limits our ability to determine whether the observed biomarker changes are specific to AF or linked to other pathophysiological processes. Future studies should incorporate appropriate control groups without AF or leverage Mendelian randomization analyses to improve our understanding in the pathophysiology of AF.

84% of the subjects were anticoagulated. Were results for stroke or bleeding different between coagulated and non-coagulated patients?

Response: The main reason why some AF patients are not anticoagulated is due to their low risk of stroke or high bleeding risk, such that there is a high risk of bias in analyzing only patients not on anticoagulation. Furthermore, only 605 patients (16%) were not receiving oral anticoagulation at baseline in our cohort, and the number of events was small (44 major bleeds, 32 all strokes, and 28 ischemic strokes). We therefore decided to conduct a sensitivity analysis including only patients who were on oral anticoagulation at baseline (n=3,212). The associations between biomarkers and major bleeding, all strokes, and ischemic stroke remained largely consistent. The results are shown below and in the Supplement.

Supplementary Table 11. Sensitivity analysis of biomarkers and risk of major bleeding in patients on oral anticoagulation (n=3,212)

Biomarker	Age and sex adjusted model		Multivariable model		Combined model (backward selection)	
	HR (95% CI)*	P value	HR (95% CI)	P value	HR (95% CI)	P value
ANG-2	1.21 (1.09-1.34)	0.0002	1.15 (1.03-1.28)	0.009	-	-
eGFR	0.80 (0.70-0.91)	0.0009	0.96 (0.82-1.13)	0.66	-	-
Cystatin C	1.32 (1.21-1.44)	9.2x10 ⁻¹⁰	1.24 (1.10-1.38)	0.0002	-	-
D-dimer	1.17 (1.07-1.28)	0.0004	1.13 (1.03-1.24)	0.009	-	-
ALAT	0.93 (0.83-1.04)	0.19	0.95 (0.85-1.06)	0.34	-	-
GDF-15	1.47 (1.32-1.65)	1.1x10 ⁻¹¹	1.40 (1.23-1.60)	5.5x10 ⁻⁷	1.21 (1.03-1.41)	0.0217
Hs-CRP	1.14 (1.04-1.25)	0.007	1.11 (1.01-1.23)	0.03	-	-
IGFBP-7	1.39 (1.26-1.54)	1.3x10 ⁻¹⁰	1.29 (1.15-1.45)	1.1x10 ⁻⁵	1.11 (0.96-1.27)	0.14
IL-6	1.26 (1.15-1.38)	1.1x10 ⁻⁶	1.22 (1.11-1.34)	6.5x10 ⁻⁵	1.12 (1.00-1.24)	0.0439
NT-proBNP	1.31 (1.15-1.49)	4.7x10 ⁻⁵	1.19 (1.04-1.37)	0.011	-	-
OPN	1.40 (1.27-1.54)	3.3x10 ⁻¹²	1.32 (1.18-1.47)	1.4x10 ⁻⁶	-	-
hsTropT	1.38 (1.25-1.53)	2.3x10 ⁻¹⁰	1.31 (1.17-1.46)	3.3x10 ⁻⁶	1.15 (1.00-1.31)	0.0411

*Hazard ratios are standardized per 1-SD increase in biomarker. Multivariable models are adjusted for additional factors controlled for body mass index, current smoker, systolic blood pressure, history of diabetes, prior stroke or TIA, history of heart failure, chronic kidney disease and coronary artery disease. The combined model is adjusted for covariates from the multivariable model plus all significant biomarkers after backward selection based on Akaike Information Criterion (AIC).

Supplementary Table 12. Sensitivity analysis of biomarkers and risk of ischemic stroke in patients on oral anticoagulation (n=3,212)

Biomarker	Age and sex adjusted model		Multivariable model		Combined model (backward selection)	
	HR (95% CI)*	P value	HR (95% CI)	P value	HR (95% CI)	P value
ANG-2	1.33 (1.15-1.55)	0.00015	1.27 (1.08-1.49)	0.00326	-	-
eGFR	0.98 (0.80-1.20)	0.86	1.09 (0.88-1.35)	0.41	-	-
Cystatin C	1.13 (0.95-1.33)	0.16	1.06 (0.86-1.29)	0.60	-	-
D-dimer	1.12 (0.97-1.30)	0.12	1.12 (0.96-1.30)	0.16	-	-
ALAT	0.87 (0.73-1.03)	0.10	0.86 (0.73-1.02)	0.09	0.83 (0.70-0.99)	0.0338
GDF-15	1.21 (1.01-1.45)	0.037	1.09 (0.87-1.35)	0.46	-	-
Hs-CRP	1.07 (0.92-1.25)	0.35	1.06 (0.91-1.23)	0.48	-	-
IGFBP-7	1.16 (0.98-1.38)	0.09	1.06 (0.88-1.29)	0.53	-	-
IL-6	1.21 (1.04-1.40)	0.012	1.15 (0.99-1.35)	0.07	-	-
NT-proBNP	1.62 (1.33-1.98)	1.8x10 ⁻⁶	1.61 (1.29-2.00)	2.2x10 ⁻⁵	1.73 (1.37-2.19)	5.0x10 ⁻⁶
OPN	1.08 (0.90-1.29)	0.43	0.99 (0.81-1.23)	0.98	0.84 (0.67-1.04)	0.11
hsTropT	1.31 (1.11-1.54)	0.00115	1.26 (1.05-1.52)	0.0140	-	-

*Hazard ratios are standardized per 1-SD increase in biomarker. Multivariable models are adjusted for additional factors controlled for body mass index, current smoker, systolic blood pressure, history of diabetes, prior stroke or TIA, history of heart failure, chronic kidney disease and coronary artery disease. The combined model is adjusted for covariates from the multivariable model plus all significant biomarkers after backward selection based on Akaike Information Criterion (AIC).

Supplementary Table 13. Sensitivity analysis of biomarkers and risk of all strokes in patients on oral anticoagulation (n=3,212)

	Age and sex adjusted model		Multivariable model		Combined model (backward selection)	
Biomarker	HR (95% CI)*	P value	HR (95% CI)	P value	HR (95% CI)	P value
ANG-2	1.27 (1.11-1.46)	0.0004	1.22 (1.06-1.41)	0.005	-	-
eGFR	0.98 (0.82-1.18)	0.86	1.07 (0.88-1.30)	0.52	-	-
Cystatin C	1.11 (0.96-1.29)	0.15	1.08 (0.90-1.29)	0.40	-	-
D-dimer	1.13 (0.99-1.28)	0.07	1.12 (0.98-1.28)	0.09	-	-
ALAT	0.87 (0.75-1.02)	0.08	0.87 (0.74-1.01)	0.07	0.85 (0.73-0.98)	0.030
GDF-15	1.16 (0.98-1.37)	0.08	1.08 (0.89-1.31)	0.44	-	-
Hs-CRP	1.04 (0.91-1.19)	0.58	1.03 (0.90-1.18)	0.67	-	-
IGFBP-7	1.16 (0.99-1.35)	0.06	1.09 (0.92-1.29)	0.31	-	-
IL-6	1.21 (1.06-1.37)	0.0047	1.17 (1.02-1.34)	0.0272	1.11 (0.95-1.29)	0.19
NT-proBNP	1.56 (1.33-1.90)	2.6×10^{-7}	1.59 (1.31-1.93)	2.8×10^{-6}	1.62 (1.32-2.00)	5.1×10^{-6}
OPN	1.11 (0.96-1.30)	0.17	1.08 (0.90-1.30)	0.40	0.89 (0.73-1.09)	0.25
hsTropT	1.30 (1.13-1.50)	0.0003	1.29 (1.09-1.51)	0.00218	-	-

*Hazard ratios are standardized per 1-SD increase in biomarker. Multivariable models are adjusted for additional factors controlled for body mass index, current smoker, systolic blood pressure, history of diabetes, prior stroke or TIA, history of heart failure, chronic kidney disease and coronary artery disease. The combined model is adjusted for covariates from the multivariable model plus all significant biomarkers after backward selection based on Akaike Information Criterion (AIC).

Supplementary Table 15. Discriminative ability of Cox and machine-learning models for outcomes with and without biomarkers of patients on oral anticoagulation (n=3,212)

Outcomes	Model	AUC_{Base} (95% CI)	AUC_{Base+biomarkers} (95% CI)	P value
Major bleeding	Combined Cox model	0.65 (0.62-0.68)	0.67 (0.65-0.70)	0.01212
	LASSO	0.67 (0.64-0.71)	0.68 (0.64-0.71)	0.8446
	Random forest	0.61 (0.57-0.65)	0.63 (0.59-0.66)	0.2568
	XGBoost	0.95 (0.93-0.96)	0.98 (0.97-0.99)	0.0001649
Ischemic stroke	Combined Cox model	0.65 (0.61-0.69)	0.69 (0.65-0.73)	0.005113
	LASSO	0.69 (0.64-0.73)	0.70 (0.65-0.75)	0.6453
	Random forest	0.58 (0.53-0.64)	0.59 (0.54-0.64)	0.6897
	XGBoost	0.97 (0.96-0.99)	0.98 (0.96-0.99)	0.5465
Any stroke	Combined Cox model	0.66 (0.62-0.70)	0.69 (0.66-0.73)	0.002828
	LASSO	0.69 (0.65-0.74)	0.71 (0.66-0.75)	0.7215
	Random forest	0.60 (0.55-0.65)	0.61 (0.56-0.65)	0.8688
	XGBoost	0.96 (0.95-0.98)	0.99 (0.98-0.99)	0.0003617

Cox base model includes age, sex, body mass index, current smoker, systolic blood pressure, history of diabetes, prior stroke or TIA, history of heart failure, chronic kidney disease and coronary artery disease.
Base models from machine learning models (LASSO, Random forest, XGBoost) includes all variables listed in the Supplementary Table 2.

Supplementary Figure 3. Associations between selected biomarkers and major bleeding, ischemic stroke and any stroke from combined Cox models of patients on oral anticoagulation (n=3,212)

Supplementary Figure 4. Relative importance of predictors from combined Cox models for major bleeding, ischemic stroke and any stroke of patients on oral anticoagulation (n=3,212)

Supplementary Figure 6. Predictive performance of Cox and machine learning models for major bleeding, ischemic stroke and any stroke with and without biomarkers of patients on oral anticoagulation (n=3,212)

We added the following statement in the methods section on page 27:

“To assess the consistency of our findings, we conducted a sensitivity analysis for stroke and major bleeding, restricting the cohort to patients who were on oral anticoagulation therapy at baseline.”

We included a statement in the results section on page 6:

“In sensitivity analyses restricted to patients on oral anticoagulation at baseline, the associations between biomarkers and major bleeding, all strokes and ischemic stroke remained consistent (Supplementary Tables 11-13 and Supplementary Figures 3-4).”

As well as a statement on page 7:

“In sensitivity analyses restricted to patients receiving oral anticoagulation, the results for stroke and major bleeding remained consistent, with most models incorporating biomarkers showing higher AUC values (Supplementary Table 15 and Supplementary Figure 6).”

And added a sentence in the discussion on pages 11-12:

“Lastly, 16% of patients were not on oral anticoagulation, which may have influenced the associations between biomarkers and some outcomes. However, we showed that the results were consistent when analyses were restricted to those on anticoagulation.”

How were the biomarkers selected? I was surprised to see BMP10 neither included in the analysis nor mentioned in the discussion section

(<https://doi.org/10.1093/eurheartj/ehae611>, <https://doi.org/10.1038/s41598-023-42331-7>).

Response: As outlined in our response to reviewer #1, the panel of 12 biomarkers was selected based on extensive literature review and prior evidence linking them to key pathophysiological pathways implicated in AF and its complications. Please see our extensive response to reviewer #1 on page 9 above. Regarding BMP10, this biomarker was measured only in a subset of the Swiss-AF cohort for experimental purposes and would have significantly reduced sample size and power if we had included it. We have now addressed the potential implications of BMP10, particularly its associations with ischemic stroke and adverse outcome events in AF patients, in the discussion on page 9:

“Bone morphogenetic protein 10 (BMP10) has been associated with incident AF in a population free of AF at baseline²². Moreover, among patients with established AF, elevated BMP10 levels have been linked to a higher risk of ischemic stroke

independent of oral anticoagulation treatment²³, and an increased incidence of adverse outcome events compared to those with lower levels²⁴. Future studies should investigate whether a more comprehensive proteomic analysis can provide deeper insights into the pathophysiology of AF complications and enhance risk stratification strategies.”

We added the suggested reference along with additional citations:

22 Chua, W. et al. An angiotensin 2, FGF23, and BMP10 biomarker signature differentiates atrial fibrillation from other concomitant cardiovascular conditions. *Sci Rep* 13, 16743 (2023).

23 Hijazi, Z. et al. Bone morphogenetic protein 10: a novel risk marker of ischaemic stroke in patients with atrial fibrillation. *Eur Heart J* 44, 208-218 (2023).

24 Hennings, E. et al. Bone Morphogenetic Protein 10-A Novel Biomarker to Predict Adverse Outcomes in Patients With Atrial Fibrillation. *J Am Heart Assoc* 12, e028255 (2023).

As outlined in the response to reviewer #1, we added the following statement in the limitations section in page 11:

“Fourth, while our panel of biomarkers was chosen based on a thorough literature review and strong evidence linking them to AF pathophysiology and its complications, we acknowledge that any selection process is somewhat arbitrary and may miss relevant biomarker associations.”

The study demonstrates a modest but statistically significant added value of biomarkers for complication risk prediction. In my opinion a statement of the strength of the effect (modest) should be made as well wherever a statement about the statistical significance is made.

Response: We appreciate this observation and have revised the manuscript to explicitly state that the improvements in predictive power are statistically significant yet modest in magnitude.

We softened our tone and rephrased the following statements in the discussion section:

Page 8: “The integration of these biomarkers into both traditional and machine learning-based predictive models significantly improved risk prediction, providing a more comprehensive assessment of adverse cardiovascular outcomes in this population. The improvement in predictive power was modest for most analyses.”

Page 9: “We demonstrated that incorporating key biomarkers into prediction models led to a modest but significant improvement in the discriminatory ability of Cox and most machine learning models.”

Page 11: "In our study, we demonstrated that machine learning models, such as XGBoost and LASSO, achieved modest but significant improvements in predictive performance when biomarkers were included."

Page 12: "By integrating these biomarkers into both traditional and machine learning-based risk models, we enhanced predictive accuracy, underscoring the potential clinical utility of biomarker-informed risk assessments in refining and optimizing the management in patients with AF."

Minor points:

- line 66: "less invasive" compared to which other methods?

Response: By "less invasive," we are referring to the comparison with more invasive methods, such as tissue biopsies or imaging procedures, which often require more significant patient intervention and are associated with higher risks and costs. Circulating biomarkers offer a less invasive alternative for assessing pathological pathways in individual patients. We rephrased the sentence on page 4 as follows:

"Circulating biomarkers that reflect these pathological pathways offer a promising and less invasive approach to assessing their involvement in individual patients, compared to more invasive diagnostic methods, such as tissue biopsies."

- line 89: the composite outcome should be defined in the main text

Response: We agree and rephrased the following sentence on page 5 as follows:

"For the composite outcome of cardiovascular death, nonfatal ischemic stroke, nonfatal systemic embolism, or nonfatal myocardial infarction, 5 biomarkers including d-dimer, GDF-15, IL-6, NT-proBNP and hsTropT independently contributed to the model fit (Supplementary Table 1, Figure 1A)."

- line 153: please double check the sentence ("surpassing than")

Response: We have rephrased the sentence for clarity on page 8 as follows.

"Our findings not only confirm this association but also highlight GDF-15 as a robust predictor of heart failure hospitalization, with a predictive strength comparable to NT-proBNP and exceeding that of IGFBP-7 (Supplementary Table 2)¹¹."

I appreciate that the code is available. Ideally it would be persistently archived, for example via the GitHub2Zenodo integration.

Response: As requested, we have archived our code using the GitHub–Zenodo integration. The revised manuscript now includes the corresponding DOI (<https://doi.org/10.5281/zenodo.14977544>) to ensure persistent accessibility. We updated the Code Availability statement on page 13 as follows:

“The code is publicly available via GitHub–Zenodo and can be accessed using the DOI: <https://doi.org/10.5281/zenodo.14977544>.”

The imputation script appears empty.

Response: We have updated and verified the imputation script to ensure that it contains the complete and functional code. The updated version is now available in the repository. We apologize for the inconvenience.

The other scripts seem solid at first glance except for some hard coded paths and the very long files without dedicated functions.

Response: We have refactored our code to remove hard-coded paths and modularized lengthy scripts into dedicated functions to enhance readability and maintainability. We appreciate the reviewer’s suggestions, which have helped improve our codebase.

Finally, we again appreciated the thoughtful review of our work.

Thank you for considering our revised version.

With best wishes,

Pascal B. Meyre, MD PhD

Reviewer #1:

The authors appropriately corrected the manuscript. It is now acceptable for publication in Nature Communications.

Response: We thank the reviewer for his positive assessment and support of our work.

Reviewer #3:

The authors have improved the manuscript significantly and addressed all my comments sufficiently.

Response: We are grateful for the reviewer's positive feedback and the valuable suggestions provided during the review process.

When looking at the code repository, it seems that the latest versions including the refactoring mentioned in the response to my comment has not been uploaded yet. Hard-coded paths and very long, rather unstructured files remain.

Response: We thank the reviewer for highlighting this important issue. We have now updated the code repository to include the latest version of the analysis scripts, as described in our previous response. These revisions eliminate hard-coded paths wherever feasible, and restructure the scripts to enhance clarity and reproducibility. The updated repository can be accessed at: <https://doi.org/10.5281/zenodo.15653551>.

We sincerely thank the Editors and Reviewers for their time and constructive input, which have significantly improved our manuscript. We look forward to the possibility of publication in *Nature Communications*.

With best wishes,

Pascal B. Meyre, MD, PhD